# Research on complex dynamic behavior of automakers based on complex system theory

**Jinhuan Tang**[◉], **Xinying Si**[◉], **Qiong Wu**[iD]*[◉], **Xiangchen Li**

School of Economics and Management, Shenyang Aerospace University, Shenyang, China

◉ These authors contributed equally to this work.
* 18447071705@163.com

## Abstract

"Prevent minor issues before they become major problems, and prepare for the future." This study utilizes complex system theory to introduce a nonlinear dynamic system for examining the production and emission reduction strategies of new energy vehicle (NEV) and gasoline vehicle (GV) manufacturers under the dual credit (DC) policy over a long-term game process. By considering production delays, we analyze dynamic behaviors within a duopoly automotive system, including stable regions, bifurcation, chaotic attractors, and the Largest Lyapunov exponent (LLE). The results show that: (1) As production and carbon emission adjustment parameters increase, the decision-making system for both automakers can slip into disorder, posing a risk of disruption within the automotive industry. (2) In stable regions, GVs' carbon emission adjustments do not affect the production of either NEVs or GVs, while NEVs demonstrate greater flexibility in production adjustments compared to GVs. (3) The industry system will likely benefit from delay production decisions that could help stabilize the automobile market. The study provides theoretical support for the smooth transformation of old and new driving forces in the automobile industry.

## 1. Introduction

China's automotive industry, propelled by strong national policies, has seen a continuous rise in production and sales volumes, securing a leading position globally. There were 281 million vehicles in China by 2020, an average of 201 per 1,000 people. China has nearly twice the number of cars sold annually compared to the United States [1]. However, with more than 800 vehicles per 1,000 people, the US remains the country with the largest vehicle ownership in the world [2]. This means that there is still a significant potential demand for vehicles in China. Unfortunately, China is also the world's largest energy consumer and carbon emitter. The transportation sector accounts for approximately 10% of China's total carbon emissions, with road transportation accounting for approximately 80% of these transport-related carbon emissions [3–6]. To achieve China's carbon neutrality goal, substantial carbon reductions, decarbonization, and a green transformation in the automotive industry are essential [7,8]. As part of this transformation, the auto industry has been moving away from gas vehicles (GVs) and towards the development of new energy vehicles (NEVs). Given their potential to save energy

**Funding:** This study was funded by the Liaoning Province Social Science Planning Fund Project

(Grant Number: L19CGL008), with Professor Jinhuan Tang as the principal investigator.

**Competing interests:** The authors have declared that no competing interests exist.

and be more environment-friendly, NEVs also have the advantage of pollution-free emissions when using clean energy [9,10]. In 2021, Chinese automobile sales reached 26.275 million, 3.8% year-on-year, of which NEVs completed 3.521 million. China has also had the highest NEV sales in the world for seven consecutive years. However, at 13.4%, the market penetration rate shows there is still a huge potential for the development of NEVs [11].

In the context of the "double carbon" goal, the Chinese government has enacted a series of regulations to support the auto industry as it transitions toward environmental sustainability [12,13]. In 2017, the government introduced the "Parallel Management Measures for Average Fuel Consumption of Passenger Cars and NEV Credits" policy, which is widely known as the Dual Credit (DC) policy. This policy seeks to meet carbon emission reduction targets by encouraging automobile manufacturers to enhance the fuel efficiency of gasoline-powered vehicles and increase the production share of NEVs. The implementation of this policy marks the transition of the NEV industry from being policy-driven to being market-driven. This can drive the auto market and force traditional energy vehicle enterprises to transition towards the production of NEVs [14]. For instance, since it stopped producing GVs in March 2022, BYD Co. Ltd. has become the first traditional vehicle enterprise in the world to officially transition to exclusively producing NEVs [15]. Currently, the auto market comprises automakers that only produce GVs (Corvette, Jeep, and Rolls Royce), focus exclusively on NEVs (Tesla, XPENG, and NIO), and those that produce both GVs and NEVs (Mazda, Geely, and Great Wall). Interestingly, some GVs and NEVs share similar appearances and configurations, with their main differences lying in energy consumption, emissions, sale prices, and endurance mileage. Taking the Audi Q5 and Tesla Model Y as examples, both models reflect the refinement and fashion sense of high-end vehicles in terms of lighting, seating, and aesthetic design. However, the 2.0T version of the Audi Q5 has an average fuel consumption of about 8 liters per 100 kilometers. According to the calculation that each liter of gasoline produces 2.3 kilograms of carbon dioxide emissions, it is approximately 18.4 kilograms per 100 kilometers, which is 2.5 times that of the Tesla Model Y. As the DC policy becomes more stringent, this study focuses on how GV automakers can make effective carbon emission reduction decisions.

Traditional oligopolistic competition models, such as the Cournot and Bertrand models, are mostly based on the assumption of complete rationality and complete information to study linear dynamic equilibrium, and this has certain limitations when dealing with complex and volatile real-world problems. The complex system theory breaks the traditional linear causal relationship, explains the complex, dynamic and nonlinear characteristics of things. As a branch of this theory, nonlinear dynamics can reveal some dynamical behaviors of the system. The long-term repeated game model with bounded rationality based on the analysis framework of nonlinear dynamic system has specific advantages. The model relaxes the assumption of complete rationality and information and can better capture the complex phenomena of market dynamic evolution. Therefore, this study establishes a competitive model with a finite rational duopoly of automakers in the auto market and considers the impact of delay decisions on automakers' complex dynamics under the DC policy. The core issue is to explore how the government intervenes in the stable growth of the auto industry and how NEV and GV automakers make decisions that will be beneficial to their long-term stable development.

This study aims to answer the following key questions: (1) Will the carbon reduction adjustment strategies of GV affect the production of both vehicle types? (2) Do automakers with competitive relationships consider competitor's strategies when making decisions? (3) How do delay production decisions affect automakers' strategy adjustments? (4) What is the impact of the simultaneous decision adjustments of both types of automakers on the stability of the auto market? (5) How can automakers maintain the automotive market under government regulations?

To solve these problems, this study focuses on duopoly automakers that produce NEVs and GVs. We establish a long-term repeated game model and analyze the equilibrium strategies of automakers under the DC policy. The main contributions of this study are as follows: (1) We construct a long-term repeated game model of automakers under the DC policy. This model is different from a traditional short-term game model and provides an analytical tool for portraying the dynamic game between the two types of automakers. (2) We construct a nonlinear dynamic system of game decisions among automakers by incorporating adjustment parameters of the decision variables. The stability of the system is judged by the Jury criterion, and we obtain a stability range for automakers' adjustments to production and carbon emission reductions, thus providing a guide for government policy making. (3) We analyze the dynamic game equilibrium strategy of duopoly automakers and compared the two cases of delay and no-delay decisions. We find that delay decisions can expand the stable range of the system and allow for more flexible and effective adjustment decisions.

The rest is structured as follows: Section 2 reviews the relevant research. Section 3 establishes a long-term repeated game model for two duopoly automakers. In Section 4, the parameter setting is given, and the numerical simulation and discussion are carried out. The main conclusions are drawn, then some management implications and policy recommendations are put forward in Section 5.

## 2. Literature review

The research highly relevant to this paper mainly involves two streams: the development strategy of the auto industry under the DC policy and supply chain decision under long-term repeated game.

### 2.1. Research on the DC policy regime

Scholars mainly focus on decision-making optimization of auto industry under the DC policy, mostly from the perspective of automakers and supply chain. From the point of automakers, Lou et al. [16] developed a decision model and discussed the impact of the DC policy on enterprise decision-making. Yang et al. [17] made a comparison of the government pricing model and market pricing model for the DC transaction price through optimization theory. He et al. [18] developed a net present value model and analyzed the optimal timing of electric transformation of automakers under DC policy. The above research study the optimal decision of a single automaker, but have not considered the impact of competition between automakers in the market on the decision.

Subsequently, some scholars involved the competitive relationship between heterogeneous automakers in their studies and explored the optimal decisions of two automakers producing NEVs and GVs, respectively. For example, Cheng et al. [19] proposed a benchmark competition model and three coopetition models, they examined the optimal strategies of both parties in a stable credit market and with credit trading risk. Zhang et al. [20] studied the optimal decision and social welfare of two automakers with different market power. Some scholars studied the pricing and emission reduction decisions of two automakers, one producing NEVs and GVs, the other one only NEVs [21]. On this basis, Li et al. [22] analyzed the impact of subsidy policy and the DC policy on the production decisions of automakers considering battery recycling in a competitive environment, where market demand was influenced by heterogeneous consumers, but ignored the cooperative relationship between automakers and their upstream or downstream enterprises.

From the point of auto supply chain, some scholars analyzed the impact of policy substitution on production and pricing strategies [23,24]. Also some researches constructed

cooperative and non-cooperative game model involving suppliers and automakers, and compared the optimal strategies of supply chain members [25–27]. Meng et al. [28] proposed an R&D decision-making model and analyzed the optimal strategy for NEV development in combination with the financial constraints of suppliers. Rao et al. [29] analyzed the disparity of the influence of the DC policy on the financial performance of upstream and downstream enterprises of NEVs by using the Interrupted Time Series method.

Some scholars considered the coopetition relationship of multiple members in auto supply chain. For example, Liu et al. [30] researched a three-level supply chain with automakers, retailers and consumers, they established a Stackelberg game model to analyze the optimal pricing of auto supply chain members. Li et al. [31] utilized a mixed integer linear programming to investigate the impacts of subsidy policy and the DC policy on NEVs and GVs production decision from an across-chain perspective. Liu et al. [32] studied the role of the government, automakers and consumers in the process of banning GV based on co-evolutionary game theory, and their research results were of great significance for the smooth delisting of GV.

In conclusion, the above literature use game theory to study the decision optimization of automakers and supply chains under the DC policy. However, there are few studies on the dynamic complexity and stability of auto market competition.

## 2.2. Research on supply chain decision under long-term repeated game

There are already existing researches that combine the game theory with the complex system theory to construct a long-term repeated game model of the supply chain, and analyzes the complex dynamic characteristics of the decision system. Long-term repeated game can be divided into two situations: no-delay decision and delay decisions. This section will introduce literature related to the long-term repeated game whether the supply chain considers the delay decisions.

The influence of no-delay decision on the complexity of supply chain system is reviewed, some scholars studied the dynamic competition between online and traditional retailers in a dual-channel supply chain, analyzed the stability and complexity of the equilibrium point, and provided informative insights on how to manage the relevant factors in the supply chain to reduce or even eliminate system chaos [33–36]. Bao et al. [37] constructed a long-term repeated game model of parallel supply chain composed of NEV and GV manufacturers, they studied the dynamic evolution characteristics of manufacturers by using complex dynamics theory. Xie et al. [38] and Chen et al. [39] studied the stability of the dynamic game system between manufacturers and retailers, their research results provided suggestions for maintaining the stability of the supply chain system. Matouk et al. [40] considered a heterogeneous triopoly game with an isoelastic demand, they studied complex features such as bifurcation and chaos. Fan et al. [41] incorporated the retailer's altruistic behavior into the low-carbon supply chain considering consumers' low-carbon preference, they explored the impacts of the retailer's altruism preference, consumers' low-carbon preference and decision parameters on the complex nonlinear dynamic behaviors of the two models. Zhang et al. [42] examine the impact of carbon emissions during the preservation process of fresh products on both traditional and low-carbon fresh supply chains that incorporate emission reduction practices. Meanwhile, Bera et al. [43] investigate how consumer preferences for product traceability, environmental sustainability, and branding influence optimal decision-making and pricing strategies for firms within a heterogeneous intelligent sustainable supply chain.

Other scholars have incorporated delay decisions into the long-term repeated game model and discussed the equilibrium strategy of manufacturers and retailers [44–49]. For example,

Liu et al. [44] constructed a differential game model based on product emission reduction level, low-carbon reputation and reference, they analyzed the influence of delay and reference dual effect related parameters on the optimal decision and profit of the supply chain. Li et al. [45] studied the existence of Nash equilibrium and local asymptotic stability of the supply chain, and the results showed that appropriate delay weights could expand the stability range of the system. Dai et al. [46] established a 3D discrete dynamic model with time-delay, they discussed the influence of the adjustment speed of decision variables and the period of decision delay on the system stability. Si et al. [47] constructed a time-delay differential price game model of dual-channel supply chain and discussed the influence of system stability on price evolution trend. Bao et al. [48] and Tian et al. [49] established a dynamic game model for manufacturers and retailers, they analyzed the dynamic behaviors of the system, such as stable region, bifurcation and chaos, singular attractor and the Largest Lyapunov exponent (LLE), and the research showed that delay decisions can effectively reduce risks and uncertainties of supply chain members. The above studies have explored the long-term repeated game behavior of the supply chain under bounded rational expectations using the method of complex system theory, and analyzed how to keep the supply chain system in a stable state.

In summary, the existing research has used game theory to conduct an in-depth discussion on the development of the auto industry under the DC policy [50,51], which provides a scientific methodological basis and guide for our research, but there are still some gaps: (1) The research is analyzed from a short-term game perspective on the auto industry under the DC policy mostly [52], and rarely consider the long-term repetitive game among automakers. (2) Delay decisions is of great value to improve the accuracy of production decisions, but few studies have introduced delay decisions into auto industry chain decision-making. (3) The existing literature generally analyze the optimal solution for individual profit maximization [53], but few researches discuss period-doubling bifurcation and chaos caused by the adjustment parameters of decision variables. Therefore, this paper constructs a long-term repeated game model under the DC policy, analyzes the stability and dynamic characteristics of the competitive equilibrium of automakers. We provide theoretical support for enterprise competition and market regulation for the auto industry.

## 3. Long-term repeated game model

In this section, a new production competition model for duopoly automakers is proposed. We examine the long-term repeated game behaviors in the production and emission reduction decision of automakers.

### 3.1. Problem description

A long-term repeated game model containing a NEV and a GV automaker is established under the DC policy. NEV automaker need to make production decisions, GV automaker need to make production and carbon reduction decisions. By introducing the adjustment parameters of production and carbon emission reduction, a nonlinear dynamic system is established to study the dynamic equilibrium strategies of the two automakers.

We set two cases; one is the normal model without considering delay decisions. In this model, the automakers only need to consider the effect of current production on the strategy when making production adjustment. The other is the delay production decision. In this model, when the automakers adjust quantities, not only do they consider the influence of the current output on those in the next period, but also they will refer to the changing of the production situation in the last period. This decision-making approach is called delay production

decision [48]. Since the decision is faced with risks and uncertainties, it may be necessary to refer to both previous and current strategy when making the next decision.

As we known, automakers in China must meet the DC policy. Suppose that the credits of GV and NEV should be completed within a period. the CAFC of GV automaker must be above zero in the end of any period. "Dual credit" accounting formula: CAFC credit is $(\bar{e} - e_g - \delta)q_g$ and NEV credit is $\beta q_n$, where $\bar{e}$ is the average fuel consumption value of the enterprise stipulated by the state, $e_g$ indicates the actual fuel consumption value of GV, $\delta$ is the proportion of NEV credit stipulated by the state for GV automaker, $q_n$ and $q_g$ is the production of NEV and GV respectively, and $\beta$ is the credit coefficient of NEV. The inverse demand functions of the two automakers are $p_n = a - q_n - q_g + \tau e_g$, $p_g = a - q_g - r q_n$, where $p_n$ and $p_g$ represent the market prices of NEV and GV respectively, $a$ is the market size, $\tau$ is consumers low carbon preference. The carbon emission reduction cost of GV is $C = 1/2 c_e e^2$, where $C$ presents carbon reduction investment of GV automaker, which is a one-time R&D investment, and $c_e$ is the carbon reduction investment coefficient of GV automaker, $e$ is the carbon emission reduction of GV.

According to the previous description, the profit of the two automakers is:

$$\begin{cases} \pi_n = (a - q_n - q_g + \tau e_g - c_n)q_n + \beta p_e q_n, \\ \pi_g = (a - q_g - r q_n - c_g)q_g - (e_g - \bar{e} - e + \delta)p_e q_g - \dfrac{1}{2} c_e e^2. \end{cases} \tag{1}$$

In Eq (1), $\pi_n$ and $\pi_g$ represent the profits of NEV automaker and GV automaker respectively; $c_n$ and $c_g$ indicate the production costs of NEV and GV respectively; $p_e$ is the credit transaction price.

## 3.2. Long-term repeated game model with no-delay decision

According to Eq (1), $\frac{\partial^2 \pi_n}{\partial q_n^2} = -2 < 0$, $\frac{\partial^2 \pi_g}{\partial q_g^2}\frac{\partial^2 \pi_g}{\partial e^2} - \left(\frac{\partial^2 \pi_g}{\partial q_g \partial e}\right)^2 = -p_e^2 + c_e > 0$, $\frac{\partial^2 \pi_g}{\partial q_g^2} = -2 < 0$, so there exists a unique $q_n$, $q_g$ and $e$ that maximizes the profits of the two automakers. The first-order partial derivatives of Eq (1) with respect to $q_n$, $q_g$ and $e$ are respectively as follows:

$$\begin{cases} \dfrac{\partial \pi_n}{\partial q_n} = a - c_n - 2q_n - q_g + \tau e_g + \beta p_e, \\ \dfrac{\partial \pi_g}{\partial q_g} = a - c_g - 2q_g - r q_n - p_e(e_g - \bar{e} - e + \delta), \\ \dfrac{\partial \pi_g}{\partial e} = p_e q_g - c_e e. \end{cases} \tag{2}$$

Assume that the two automakers adopt bounded rational expectations to deal with the uncertainty about production and carbon emissions reduction, they will consider the production and carbon emission reduction strategies in period $t$ when making decisions in period $t$ +1. Therefore, the bounded rational decision expression of the two automakers is as follows:

$$\begin{cases} q_n(t+1) = q_n(t) + \alpha_1 q_n(t)(a - c_n - 2q_n - q_g + \tau e_g + \beta p_e), \\ q_g(t+1) = q_g(t) + \alpha_2 q_g(t)(a - c_g - 2q_g - r q_n - p_e(e_g - \bar{e} - e + \delta)), \\ e(t+1) = e(t) + \alpha_3 e(t)(p_e q_g - c_e e), \end{cases} \tag{3}$$

where $\alpha_1$ is the production adjustment speed of NEV; $\alpha_2$ and $\alpha_3$ represent the production and carbon emission reduction adjustment speed of GV, respectively. According to Eq (3), the

dynamic game process of the two automakers can be regarded as a discrete-time nonlinear dynamic system. The system can operate stably depends on whether there is an equilibrium point that makes the two automakers keep a stable production decision. So we need to analyze the system whether there is an asymptotically stable equilibrium point. Set $q_n(t+1) = q_n(t)$, $q_g(t+1) = q_g(t)$, $e(t+1) = e(t)$ to obtain six equilibrium points of the system:

$$E_1 = (0,0,0), E_2 = \left(\frac{c_e(2B-A)-p_e^2A}{C}, \frac{c_e(2A-rB)}{C}, \frac{p_e(2A-rB)}{C}\right), E_3 = \left(0, \frac{c_eA}{2c_e-p_e^2}, \frac{p_eA}{2c_e-p_e^2}\right),$$

$$E_4 = \left(\frac{B}{2}, 0, 0\right), E_5 = \left(\frac{2B-A}{4-r}, \frac{2A-rB}{4-r}, 0\right), E_6 = \left(0, \frac{A}{2}, 0\right),$$

where $A = a - c_g - e_g p_e + \bar{e} p_e - \delta p_e$, $B = a - c_n + \tau e_g + \beta p_e$, $C = c_e(4-r) - 2p_e^2$.

Observe the above six equilibrium points: $E_1$ is the origin, $E_3$, $E_4$, $E_5$ and $E_6$ are saddle points, these five points are unstable, and $E_2$ are the only Nash equilibrium point of the system (3). The stability of the equilibrium point is determined by the eigenvalues of the Jacobian matrix. Considering the complexity of solving the characteristic equations, the stability of the system is analyzed by Jury stability criterion [36]. The expression of the Jacobian matrix $J(E)$ is as follows:

$$J(E) = \begin{bmatrix} J_{11} & -\alpha_1 q_n & 0 \\ -\alpha_2 q_g r & J_{22} & \alpha_2 p_e q_g \\ 0 & \alpha_3 e p_e & J_{33} \end{bmatrix}, \tag{4}$$

where

$$\begin{cases} J_{11} = 1 + \alpha_1(a - c_n - 4q_n - q_g + \tau e_g + \beta p_e), \\ J_{22} = 1 + \alpha_2(a - c_g - 4q_g - rq_n - p_e(e_g - \bar{e} - e + \delta)), \\ J_{33} = 1 + \alpha_3(p_e q_g - 2c_e e). \end{cases}$$

The characteristic polynomial of the Jacobian matrix is as follows:

$$F(Z) = A_3 Z^3 + A_2 Z^2 + A_1 Z + A_0. \tag{5}$$

According to the Jury criterion, the system can be stable if and only if all the coefficients of the characteristic polynomial satisfy the following conditions.

$$\begin{cases} A_0 + A_1 + A_2 + A_3 > 0, \\ A_0 - A_1 + A_2 - A_3 < 0, \\ |A_0| < A_3, \\ |A_0^2 - A_3^2| > |A_0 A_2 - A_1 A_3|. \end{cases} \tag{6}$$

### 3.3. Long-term repeated game model with delay production decisions

Since decisions are faced with risks and uncertainties, to reduce losses as much as possible, this paper assumes that the two automakers have delay production considering. When adjusting the production of $t+1$ period, automakers will refer to the change of the production situation of $t-1$ and $t$ period. The production expressions of NEV and GV under delay production decisions are as follows:

$$\begin{cases} q_n^d = \varphi_1 q_n(t) + (1 - \varphi_1)q_n(t-1), \\ q_g^d = \varphi_2 q_g(t) + (1 - \varphi_2)q_g(t-1), \end{cases} \tag{7}$$

where $q_n^d$ and $q_g^d$ represent the weighted combination of production in period $t$ and $t-1$ of NEV

and GV automaker, respectively; $\varphi_1$ and $\varphi_2$ are the weight factors, which represent the reference degree of decision in period $t$ by NEV and GV automaker, respectively; the range of weight factor is [0,1] [48]. When $\varphi_1 = \varphi_2 = 1$, the weights of production $q_n(t-1)$ and $q_g(t-1)$ are both zero. That is, the two automakers only consider production in period $t$ when making bounded rational decisions, and the situation is the same as Section 3.2; when $\varphi_1 = \varphi_2 < 1$, both automakers face delay decisions.

According to Eqs (1) and (7), the first partial derivatives of $\pi_n$ and $\pi_g$ with respect to $q_n$, $q_g$, and $e$ are respectively as follows:

$$
\begin{cases}
\dfrac{\partial \pi_n}{\partial q_n} = a - c_n - 2q_n^d(t) - q_g^d(t) + \tau e_g + \beta p_e, \\[2mm]
\dfrac{\partial \pi_g}{\partial q_g} = a - c_g - 2q_g^d(t) - rq_n^d(t) - p_e(e_g - \bar{e} - e(t) + \delta), \\[2mm]
\dfrac{\partial \pi_g}{\partial e} = p_e q_g^d(t) - c_e e.
\end{cases}
\tag{8}
$$

According to Eqs (7) and (8), we can obtain the delay production decisions system as follows:

$$
\begin{cases}
q_n(t+1) = q_n(t) + \gamma_1 q_n(t)(B - 2(\varphi_1 q_n(t) + (1-\varphi_1)q_n(t-1)) - (\varphi_2 q_g(t) + (1-\varphi_2)q_g(t-1))), \\
q_g(t+1) = q_g(t) + \gamma_2 q_g(t)(A - 2(\varphi_2 q_g(t) + (1-\varphi_2)q_g(t-1)) - r(\varphi_1 q_n(t) + (1-\varphi_1)q_n(t-1)) + p_e e(t)), \\
e(t+1) = e(t) + \gamma_3 e(t)(p_e(\varphi_2 q_g(t) + (1-\varphi_2)q_g(t-1)) - c_e e(t)),
\end{cases}
\tag{9}
$$

where $\gamma_1$, $\gamma_2$ and $\gamma_3$ represent the production adjustment speed of NEV, GV, and the carbon emission reduction adjustment speed of GV under the situation of delay decisions, respectively. For convenience, let $x_n(t+1) = q_n(t)$, $x_g(t+1) = q_g(t)$, then $x_n(t) = q_n(t-1)$, $x_g(t) = q_g(t-1)$, system (9) can be rewritten as:

$$
\begin{cases}
q_n(t+1) = q_n(t) + \gamma_1 q_n(t)(B - 2(\varphi_1 q_n(t) + (1-\varphi_1)x_n(t)) - (\varphi_2 q_g(t) + (1-\varphi_2)x_g(t))), \\
q_g(t+1) = q_g(t) + \gamma_2 q_g(t)(A - 2(\varphi_2 q_g(t) + (1-\varphi_2)x_g(t)) - r(\varphi_1 q_n(t) + (1-\varphi_1)x_n(t)) + p_e e(t)), \\
e(t+1) = e(t) + \gamma_3 e(t)(p_e(\varphi_2 q_g(t) + (1-\varphi_2)x_g(t)) - c_e e(t)), \\
x_n(t+1) = q_n(t), \\
x_g(t+1) = q_g(t).
\end{cases}
\tag{10}
$$

Let $q_n(t+1) = q_n(t)$, $q_g(t+1) = q_g(t)$, $e(t+1) = e(t)$, $x_n(t+1) = x_n(t)$, $x_g(t+1) = x_g(t)$, we can obtain the six equilibrium points of the system.

$E_1^d = (\frac{c_e(2B-A)-p_e^2 A}{C}, \frac{c_e(2A-rB)}{C}, \frac{p_e(2A-rB)}{C}, \frac{c_e(2B-A)-p_e^2 A}{C}, \frac{c_e(2A-rB)}{C})$,

$E_2^d = (\frac{B}{2}, 0, 0, \frac{B}{2}, 0)$, $E_3^d = (0, 0, 0, 0, 0)$, $E_4^d = (\frac{2B-A}{4-r}, \frac{2A-rB}{4-r}, 0, \frac{2B-A}{4-r}, \frac{2A-rB}{4-r})$,

$E_5^d = (0, \frac{c_e A}{2c_e - p_e^2}, \frac{p_e A}{2c_e - p_e^2}, 0, \frac{c_e A}{2c_e - p_e^2})$, $E_6^d = (0, \frac{A}{2}, 0, 0, \frac{A}{2})$, where $A = a - c_g - e_g p_e + \bar{e} p_e - \lambda p_e$,

$B = a - c_n + \tau e_g + \beta p_e$, $C = c_e(4-r) - 2p_e^2$. In the system (10), $E_2^d$, $E_4^d$, $E_5^d$ and $E_6^d$ are all boundary points; $E_3^d$ is the origin, $E_1^d$ is the unique equilibrium point, and it is the same as the equilibrium point in the no-delay decision model. The Jacobian matrix of system (10) can be

expressed as follows:

$$J(E^d) = \begin{bmatrix} J_{11}^d & -\gamma_1\varphi_2 q_n & 0 & -2\gamma_1(1-\varphi_1)q_n & -\gamma_1(1-\varphi_2)q_n \\ -\gamma_2\varphi_1 rq_g & J_{22}^d & \gamma_2 p_e q_g & -\gamma_2(1-\varphi_1)rq_g & -2\gamma_2(1-\varphi_2)q_g \\ 0 & \gamma_3\varphi_2 ep_e & J_{33}^d & 0 & \gamma_3(1-\varphi_2)ep_e \\ 1 & 0 & 0 & 0 & 0 \\ 0 & 1 & 0 & 0 & 0 \end{bmatrix}, \tag{11}$$

where

$$\begin{cases} J_{11}^d = 1 + \gamma_1(B - 4\varphi_1 q_n - \varphi_2 q_g - 2(1-\varphi_1)x_n - (1-\varphi_2)x_g), \\ J_{22}^d = 1 + \gamma_2(A - 4\varphi_2 q_g - r(\varphi_1 q_n + ((1-\varphi_1)x_n) + ep_e - 2(1-\varphi_2)x_g), \\ J_{33}^d = 1 + \gamma_3(p_e(\varphi_2 q_g + (1-\varphi_2)x_g) - 2c_e e). \end{cases}$$

The stability of the system is analyzed using the Jury criterion. Therefore, the Jacobian matrix must satisfy the following constraints:

$$F(Z) = A_5 Z^5 + A_4 Z^4 + A_3 Z^3 + A_2 Z^2 + A_1 Z + A_0, \tag{12}$$

$$\begin{cases} F(1) = A_0 + A_1 + A_2 + A_3 + A_4 + A_5 > 0, \\ F(-1) = A_0 - A_1 + A_2 - A_3 + A_4 - A_5 < 0, \\ |A_0| < A_5, \\ |B_0| > |B_4|, \\ |C_0| > |C_3|, \\ |D_0| > |D_2|, \end{cases}$$

$$B_0 = \begin{vmatrix} A_0 & A_5 \\ A_5 & A_0 \end{vmatrix}, B_1 = \begin{vmatrix} A_0 & A_4 \\ A_5 & A_1 \end{vmatrix}, B_2 = \begin{vmatrix} A_0 & A_3 \\ A_5 & A_2 \end{vmatrix}, B_3 = \begin{vmatrix} A_0 & A_2 \\ A_5 & A_3 \end{vmatrix}, B_4 = \begin{vmatrix} A_0 & A_1 \\ A_5 & A_4 \end{vmatrix},$$

$$C_0 = \begin{vmatrix} B_0 & B_4 \\ B_4 & B_0 \end{vmatrix}, C_1 = \begin{vmatrix} B_0 & B_3 \\ B_4 & B_1 \end{vmatrix}, C_2 = \begin{vmatrix} B_0 & B_2 \\ B_4 & B_2 \end{vmatrix}, C_3 = \begin{vmatrix} B_0 & B_1 \\ B_4 & B_3 \end{vmatrix},$$

$$D_0 = \begin{vmatrix} C_0 & C_3 \\ C_3 & C_0 \end{vmatrix}, D_1 = \begin{vmatrix} C_0 & C_2 \\ C_3 & C_1 \end{vmatrix}, D_2 = \begin{vmatrix} C_0 & C_1 \\ C_3 & C_2 \end{vmatrix}.$$

## 4. Analysis of the complex and dynamic characteristics of the system

This section conducts the numerical simulation on the model established above, and further analyzes the influence of the three adjustment parameters varying on the stability of the auto market with the help of bifurcation and chaos theory. To make the study authentic and referential, we use some open data from automakers, Ministry of Industry and Information Technology and other relevant parameters to assign values. Since it is difficult to obtain the production cost data, the production cost parameters of the two types of vehicles are designed with reference to official guiding prices. For instance, BYD, a representative enterprise of China's NEV industry, is selected as the study case. The official guiding price of its popular Qin

PLUS DM-i is 111,800 Yuan per vehicle, so $c_n = 1.1$. Meanwhile, CCAG, the most popular Chinese traditional GV enterprise, the official guiding price of its hot-selling Changan CS55PLUS is 87,900 Yuan per vehicle, so $c_g = 0.9$. According to the data from the Ministry of Industry and Information Technology, among the 93 automakers disclosed, none producing GV-only vehicles has met fuel consumption target set by the government in 2020. The actual average fuel consumption is 7.35L/100 km, the standard value is 5.81L/100 km, the NEV can get 3.32 credits per vehicle, so let $e_g = 7$, $\bar{e} = 6$, $\beta = 3$. Since the revised DC policy stipulates the NEV credit ratio is 16% in 2022, let $\delta = 0.16$. Referring to the study of Lu et al. [21] and Cheng et al. [19], let $r = 0.6$, $\tau = 0.2$, $p_e = 0.2$, $c_e = 8$. In addition, since the auto market size is large enough, the parameter should be larger than the other parameters, let $a = 20$. We employed the powerful mathematical software tools Mathematica and MATLAB for numerical simulations to create our simulation figures. These tools enabled us to accurately simulate and visualize the complex behaviors of dynamic systems, providing a robust numerical foundation for our analysis.

### 4.1. Long-term repeated game model with no-delay decision

According to the above assignments, the Nash equilibrium solution $E_2 = (6.6749, 7.4501, 0.1863)$ is obtained. After a finite number of repeated games, the system reaches a stable state at the equilibrium point. Since the Jury criterion judges the stability based on the coefficients of the characteristic polynomial of the discrete system, the coefficients in the characteristic polynomial are as follows:

$$
\begin{cases}
A_0 = -1 + 13.35\alpha_1 + 14.90\alpha_2 + 1.49\alpha_3 - 169.07\alpha_1\alpha_2 - 19.90\alpha_1\alpha_3 - 22.16\alpha_2\alpha_3 + 251.31\alpha_1\alpha_2\alpha_3, \\
A_1 = 3 - 26.70\alpha_1 - 29.80\alpha_2 - 2.98\alpha_3 + 169.07\alpha_1\alpha_2 + 19.90\alpha_1\alpha_3 + 22.16\alpha_2\alpha_3, \\
A_2 = -3 + 13.35\alpha_1 + 14.90\alpha_2 + 1.49\alpha_3, \\
A_3 = 1.
\end{cases}
$$

**4.1.1. Stability area.** Fig 1 is a 3D stability region with axes of $\alpha_1$, $\alpha_2$ and $\alpha_3$. If $(\alpha_1, \alpha_2, \alpha_3)$ is in this region, system (3) will be stable at a fixed point; otherwise, the system will be unstable,

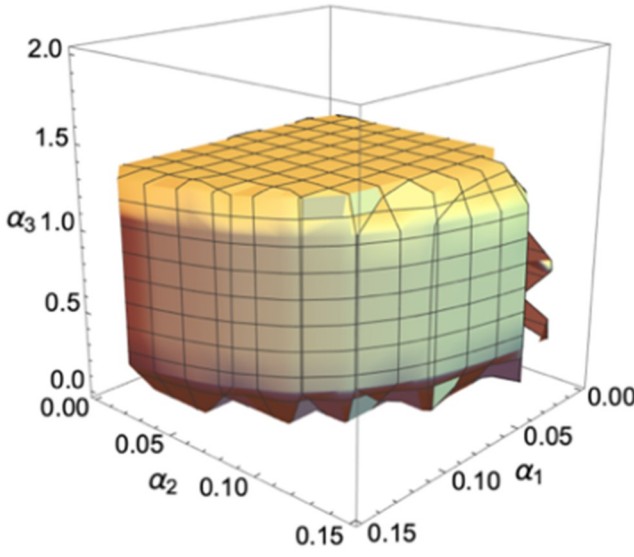

**Fig 1. The 3D stability region of the system (3).**

furthermore affecting the stability of the automaker's operation and competition. As can be seen from the Fig 1, the stability ranges, respectively, are as follows: $\alpha_1 \in (0, 0.15)$, $\alpha_2 \in (0, 0.13)$, $\alpha_3 \in (0, 1.4)$. The range of carbon emission reduction of GV is much larger than the production range of NEV and GV. With the increasing of $\alpha_1$ or $\alpha_2$, the area of the plane enclosed by the other two adjustment parameters gradually decreases; with the increasing of $\alpha_3$, the plane area enclosed by other adjustment parameters varies little, which indicates that the carbon emission reduction adjustment of GV has almost no impact on the production of NEV and GV. GV manufactures can better formulate their emission reduction strategies in combination with their enterprise strategies to meet the DC policy.

**4.1.2. Chaos bifurcation.** The Lyapunov exponent is an important quantitative tool for assessing the dynamic characteristics of a system, quantifying the average rate of convergence or divergence of neighboring trajectories in phase space over time. In particular, the LLE is crucial in revealing the chaotic nature of a system's long-term behavior. It more directly quantifies the sensitivity and chaotic characteristics of a system's dynamic actions. By observing whether the LLE is greater than zero, one can intuitively determine the system's state of dynamic chaos. A negative LLE indicates that the system is in a stable or period-doubling state; whereas an LLE of zero signifies a critical point in the system's state. Once the LLE is greater than zero, the system enters a chaotic state. Referring to the research of Zarepour et al. [54], the LLE calculation formula for discrete systems is defined as:

$$LLE = \frac{1}{N} \log(\max|\lambda \prod_{i=0}^{N-1} J_i|), \tag{13}$$

where $\lambda$ represents the eigenvalue of $J_1$, and $J_i$ is the Jacobian matrix calculated at each iteration.

Taking $\alpha_1$ as a variable, the bifurcation and LLE diagrams of system state variables $q_n$, $q_g$ and $e$ changing in the interval [0,0.18] is drawn in Fig 2. In this case, $\alpha_2$, $\alpha_3$ are 0.1 and 0.2, respectively. In Fig 2, red dots indicate the production $q_n$ of NEV, blue dots indicate the production $q_g$ of GV, black dots indicate the carbon emission reduction $e$ of GV, and the pink line represents the LLE. Therefore, the system (3) in turn remains in the stable region, period-doubling bifurcation, and chaotic state. The LLE shows the same trend as the bifurcation chart in Fig 2. By observing Fig 2A, it is found that when $\alpha_1$ increases to 0.1065, the system is at the double-cycle bifurcation state. When $\alpha_1$ increases to 0.14, the system enters a quadruple cycle bifurcation state. Moreover, Neimark-Stacker bifurcation (referred to as N-S bifurcation) appears in the chaotic state of the production adjustment parameters of NEV. Fig 2B is the local enlarged view of Fig 2A, where N-S bifurcation can be clearly observed. When $\alpha_1$ increases to 0.159, the system varies from flip bifurcation into N-S bifurcation, and the operation strategies of the two automakers are no longer in chaos but follow certain rules. We notice that the critical values of different state changes of the system can be more clearly seen from the LLE diagram, but not obvious from the bifurcation diagram.

Taking $\alpha_2$ as a variable, when $\alpha_1 = 0.1$, $\alpha_3 = 0.2$, we draw the bifurcation and LLE diagram of the system state variables $q_n$, $q_g$ and $e$ in the interval [0,0.15] in Fig 3A. In Fig 3A, pink points indicate LLE, if LLE is negative, the system is in a stable state; and if LLE is positive, the system is in a chaotic state. Similar to Fig 2A, we can observe from Fig 3A that the system eventually enters a chaotic bifurcation state with the increasing of $\alpha_2$. The bifurcation points in Fig 3A are smaller than Fig 2A, it indicates that compared with GV, the production adjustment of NEV is more flexible. Consistent with the actual situation, with the development of NEV industry, consumers' demand for NEV is increasing, and more and more enterprises choose to carry out new energy transformation.

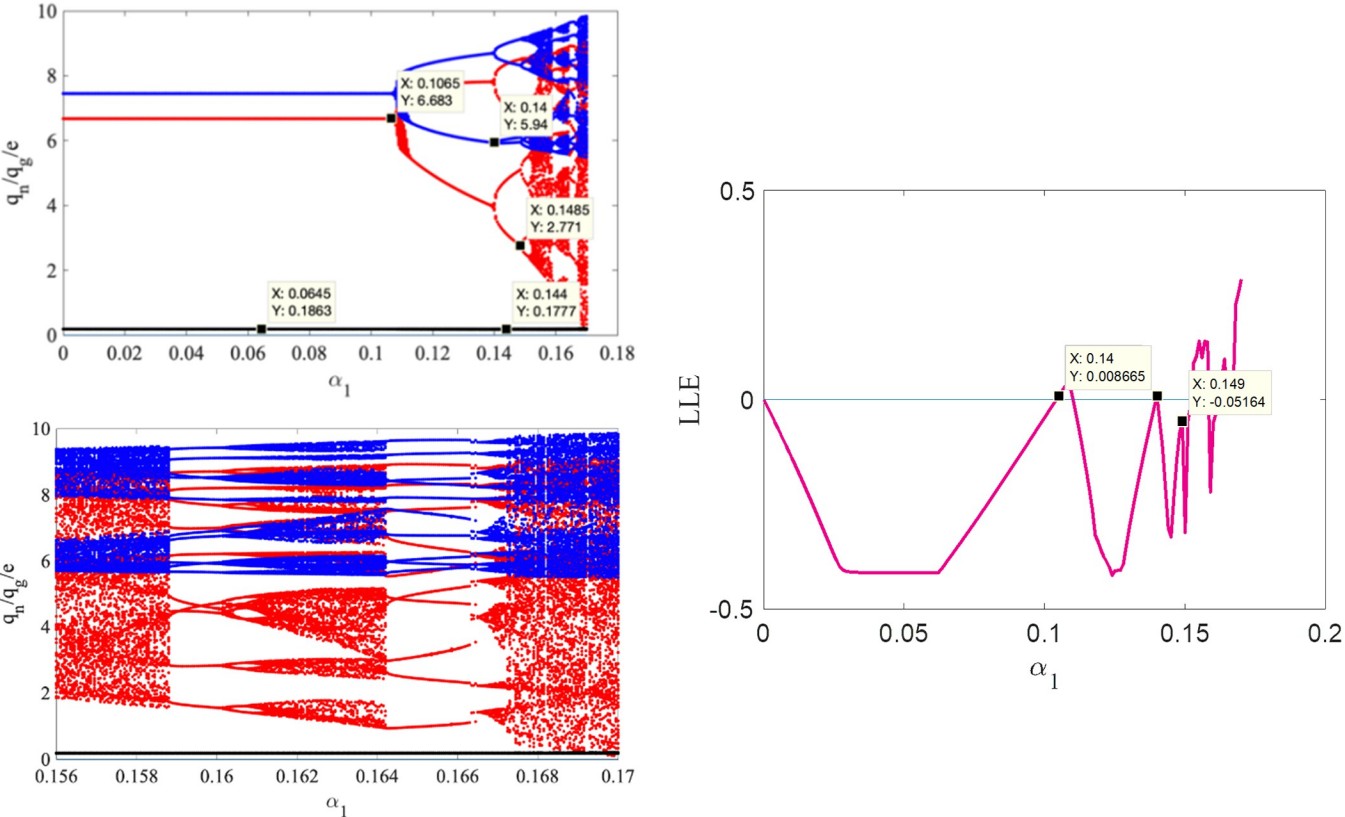

**Fig 2. Bifurcation and LLE diagrams for system (3) concerning the adjustment speed of NEV production under the conditions $\alpha_2 = 0.1$ and $\alpha_3 = 0.2$.** ((a) Bifurcation diagram with $\alpha_1 \in [0,018]$. (b) A detailed bifurcation diagram with $\alpha_1 \in [0.156, 0.17]$. (c) LLE diagram with $\alpha_1$ spanning the range $[0, 0.18]$).

Similarly, when $\alpha_1, \alpha_2$ are both 0.1, the bifurcation diagram of $\alpha_3$ changing within the interval $[0,2]$ is drawn in Fig 3B. It shows when $\alpha_3$ increases to 1.315, the system enters the double-cycle bifurcation; when $\alpha_3$ increases to 1.755, the system is at the chaotic state directly. Fig 3B again verifies the conclusion that the carbon reduction adjustment speed of GV has little impact on the production of NEV and GV.

**4.1.3. Chaos attractor.** Chaotic attractors represent the stable state of the system, and chaotic attractors do not change with the periodic state of the system. Chaotic attractors represent the trajectory of nonlinear system during long-term repeated game. When the production adjustment parameters take its values in the stable region, the chaotic attractor is a fixed point. When the production adjustment parameters are in periodic and chaotic state, the chaotic attractor will be a complex and distorted structure. Fig 4 depicts the attractors of system (3) when the production adjustment parameters are in different states. When $\alpha_1 = 0.05$, $\alpha_2 = 0.1$, and $\alpha_3 = 0.2$, the system is in a stable state. When $\alpha_1 = 0.106$, $\alpha_2 = 0.1$, and $\alpha_3 = 0.2$, the system is in a two-fold periodic bifurcation state. When $\alpha_1 = 0.106$, $\alpha_2 = 0.1$, and $\alpha_3 = 0.2$, the system is in a quadruple cycle bifurcation state. When $\alpha_1 = 0.145$, $\alpha_2 = 0.1$, and $\alpha_3 = 0.2$, the system is chaotic, and the chaotic attractor of the system is abnormal.

## 4.2. Long-term repeated game model with delay production decisions

Since the Nash equilibrium point in the delay production decisions system is the same as that in the no-delay decision system. Therefore, the coefficients of the characteristic polynomials in

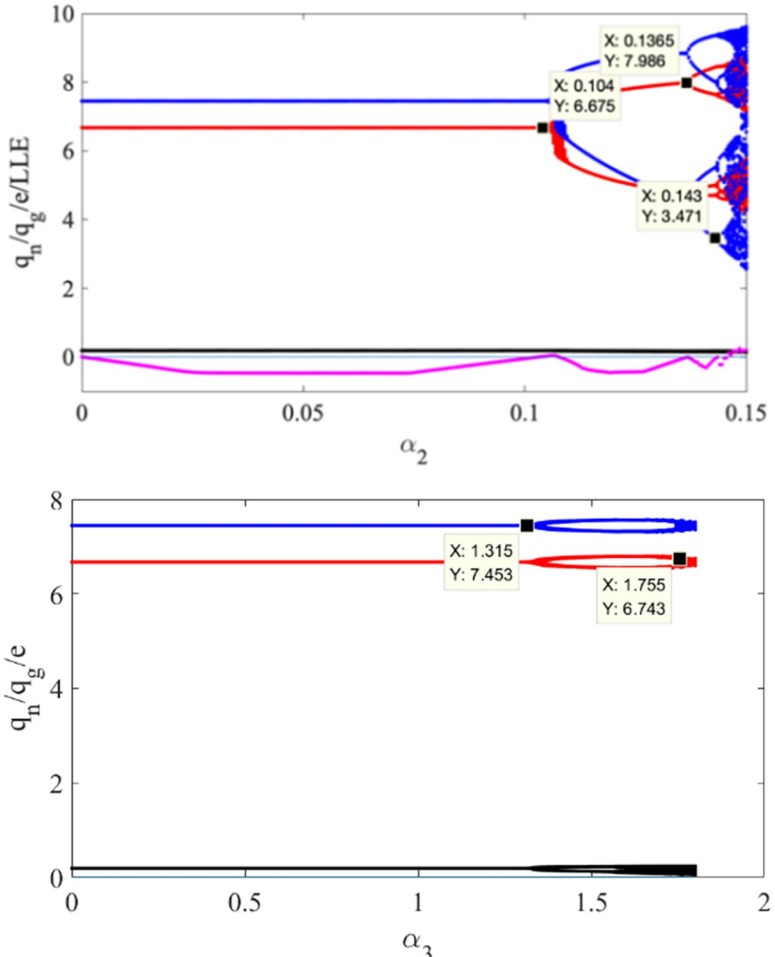

**Fig 3.** Bifurcation and LLE diagrams of the system (3) concerning the adjustment speed of GV production under the conditions $\alpha_1 = 0.1$, $\alpha_3 = 0.2$ (a), and bifurcation and LLE diagram of the system (3) concerning the regulation speed of GV carbon emission reduction under the conditions $\alpha_1 = \alpha_2 = 0.1$ (b).

this model are calculated by the above assignments as follows:

$$
\begin{cases}
A_0 = -169.08\gamma_1\gamma_2(1 - \varphi_1 - \varphi_2 + \varphi_1\varphi_2) + 251.32\gamma_1\gamma_2\gamma_3(1 - \varphi_1 - \varphi_2 + \varphi_1\varphi_2), \\
A_1 = 13.35\gamma_1(1 - \varphi_1) + 14.90\gamma_2(1 - \varphi_2) - 19.90\gamma_1\gamma_3(1 - \varphi_1) - 22.16\gamma_2\gamma_3(1 - \varphi_2) \\
\qquad + 169.08\gamma_1\gamma_2(1 - 2\varphi_1 - 2\varphi_2 + 3\varphi_1\varphi_2) + 251.32\gamma_1\gamma_2\gamma_3(\varphi_1 + \varphi_2 - 2\varphi_1\varphi_2), \\
A_2 = -1 - 13.35\gamma_1(2 - 3\varphi_1) - 29.80\gamma_2 + 44.70\gamma_2\varphi_2 + 1.49\gamma_3 + 169.08\gamma_1\gamma_2(\varphi_1 + \varphi_2 \\
\qquad - 3\varphi_1\varphi_2) + 19.90\gamma_1\gamma_3(1 - 2\varphi_1) + 22.16\gamma_3\gamma_2(1 - 2\varphi_2) + 251.32\gamma_1\gamma_2\gamma_3\varphi_1\varphi_2, \\
A_3 = 3 + 13.35\gamma_1(1 - 3\varphi_1) + 14.90\gamma_2(2 - \varphi_2) - 44.70\gamma_2\varphi_2 - 2.98\gamma_3 - 169.08\gamma_1\gamma_2(\varphi_1 \\
\qquad - 2\varphi_1\varphi_2) + 22.16\gamma_3\gamma_2\varphi_2 + 19.90\gamma_1\gamma_3\varphi_1, \\
A_4 = -3 + 1.49\gamma_3 + 13.35\gamma_1\varphi_1 + 14.90\gamma_2\varphi_2, \\
A_5 = 1.
\end{cases}
$$

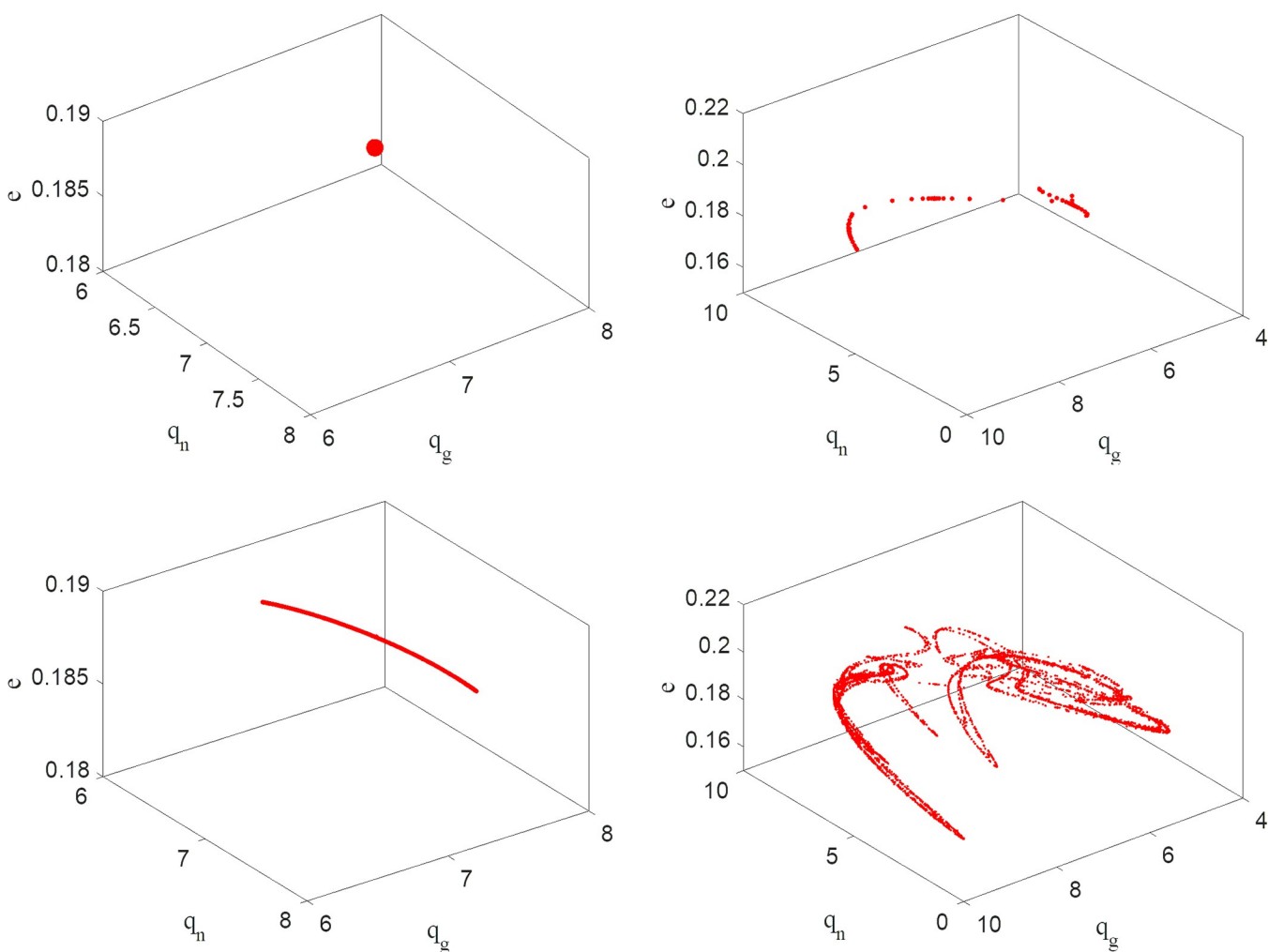

**Fig 4.** Chaotic attractors of system (3) for: a ($\alpha_1 = 0.05$, $\alpha_2 = 0.1$, $\alpha_3 = 0.2$); b ($\alpha_1 = 0.106$, $\alpha_2 = 0.1$, $\alpha_3 = 0.2$); c ($\alpha_1 = 0.145$, $\alpha_2 = 0.1$, $\alpha_3 = 0.2$); d ($\alpha_1 = 0.17$, $\alpha_2 = 0.1$, $\alpha_3 = 0.2$).

When two automakers consider the delay production decisions, their stability region is affected accordingly.

**4.2.1. Stability area.** Fig 5 shows the 3D stability region of the system (10). When both the weight factor $\varphi_1$ and $\varphi_2$ are zero, production adjustment speed of NEV is $\gamma_1 \in (0, 0.07)$, production adjustment speed for GV is $\gamma_2 \in (0, 0.04)$, carbon emission reduction adjustment speed for GV is $\gamma_3 \in (0, 1.4)$; when $\varphi_1$ and $\varphi_2$ are both 0.5, $\gamma_1 \in (0, 0.15)$, $\gamma_2 \in (0, 0.09)$, $\gamma_3 \in (0, 1.4)$; when $\varphi_1$ and $\varphi_2$ are both 0.8, $\gamma_1 \in (0, 0.25)$, $\gamma_2 \in (0, 0.22)$, $\gamma_3 \in (0, 1.4)$; When $\varphi_1$ and $\varphi_2$ are both 1, $\gamma_1 \in (0, 0.15)$, $\gamma_2 \in (0, 0.13)$, $\gamma_3 \in (0, 1.4)$, the range of $\gamma_1$, $\gamma_2$ and $\gamma_3$ is the same as the range in the no-delay situation. From the Fig 5, the carbon emission reduction does not consider the delay decisions, so the range of $\gamma_3$ does not change with the weight factor; with the increasing of $\varphi_1$ and $\varphi_2$, the range of $\gamma_1$ and $\gamma_2$ increases first and then decreases, indicating that considering delay production decisions will expand the production adjustment range of NEV and GV. Therefore, when making business strategy, the automaker should find a suitable weight factor, and make full use of delay in production in the condition of market stability, to increase the profit of the enterprise.

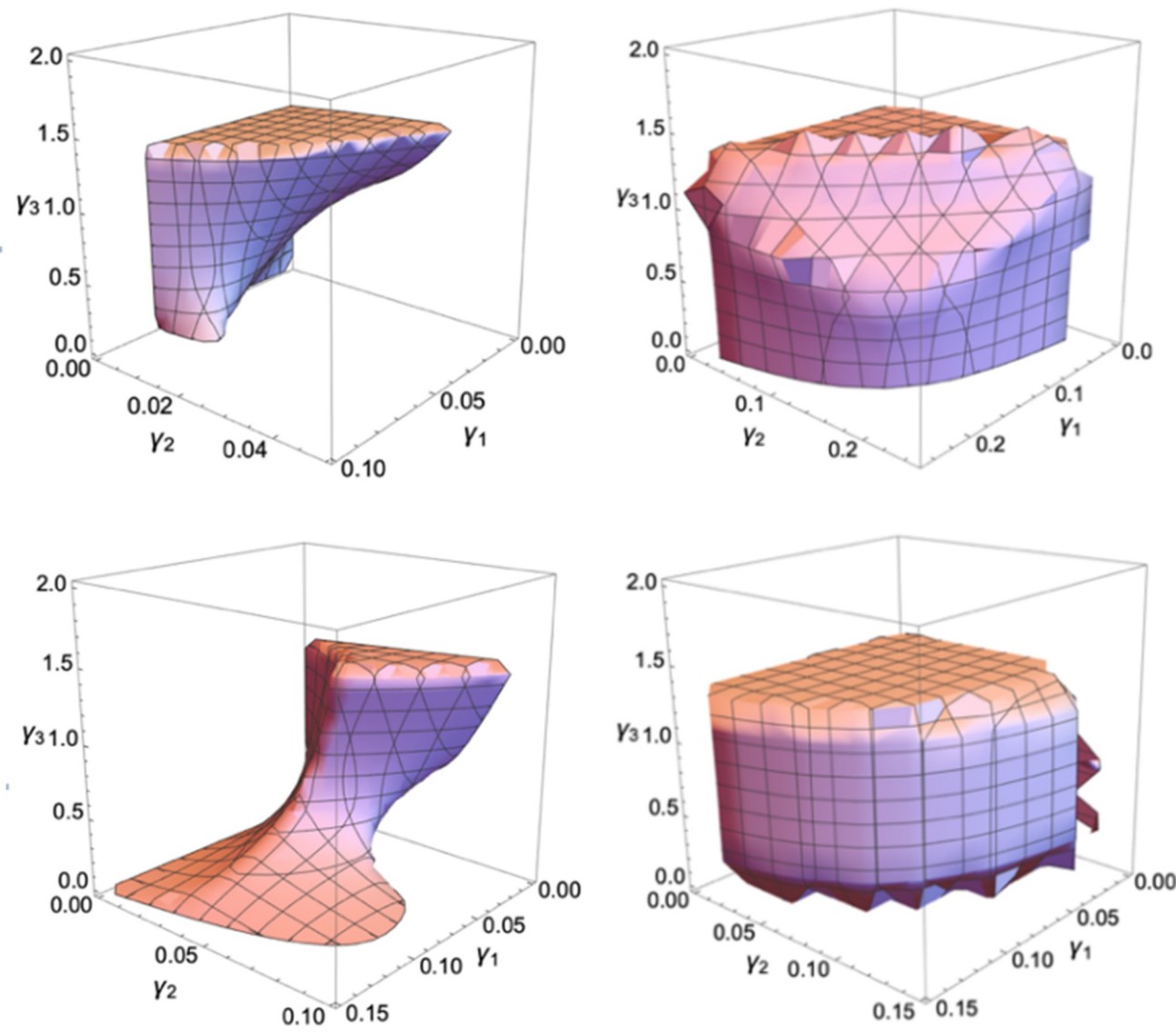

**Fig 5. The 3D stability region of the system (10) for a ($\varphi_1 = \varphi_2 = 0$); b ($\varphi_1 = \varphi_2 = 0.5$); c ($\varphi_1 = \varphi_2 = 0.8$); d ($\varphi_1 = \varphi_2 = 1$).**

**4.2.2. Chaos bifurcation.** Fig 6 shows the bifurcation diagram of the system for $\gamma_1$, when the weight factor $\varphi_1$ and $\varphi_2$ are both 0.8. From the Fig 6, when $\gamma_2$, $\gamma_3$ are 0.1 and 0.2, respectively, the system does not enter chaos state. Moreover, the value of the double-cycle bifurcation point is 0.222, which is much larger than the boundary point of the no-delay production decision case.

As shown in Fig 7, the production adjustment speed $\gamma_1$ and $\gamma_2$ are selected as the control variables. When $\gamma_3$ is 0.2, we draw the two-parameter bifurcation diagram of $\gamma_1$ and $\gamma_2$ in interval [0,0.3]. The bifurcation diagram illustrates the impact of the combined variation of the $\gamma_1$ and $\gamma_2$ on the system. When $\gamma_1$ or $\gamma_2$ below a certain threshold, system (10) remains at a stable state; when $\gamma_1$ or $\gamma_2$ increases to a value, the system will gradually enter the period-doubling bifurcation state, for example, $\gamma_1$ is 0.2, $\gamma_2$ reaches 0.2; when $\gamma_1$ and $\gamma_2$ increase to a certain value at the same time, the system will cross the period-doubling bifurcation state and enter the chaotic state directly. The synergy of the $\gamma_1$ and $\gamma_2$ has a greater impact on the stability of

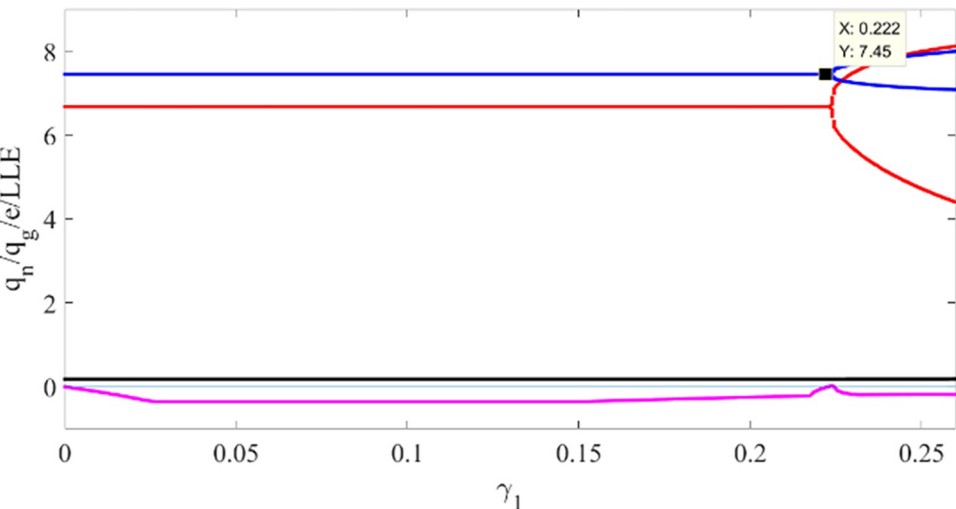

**Fig 6. Bifurcation of production adjustment parameter of NEV in System (10).**

the system. If the two automakers increase the production of both types of vehicles at the same time, the auto market will quickly become unstable.

**4.2.3. Time-series.** Fig 8 depicts the changes of NEV production in the long-term repeated game when $\gamma_1$, $\gamma_2$ and $\gamma_3$ are fixed, $\varphi_1$ and $\varphi_2$ are variable. In Fig 8A, when $\varphi_1$ and $\varphi_2$ are both zero, $q_n$ has been zero since first cycle, rose rapidly to maximum in the 52th cycles, and then disappear. Since the decision of $t+1$ period was made with complete reference to $t-1$ period, the automakers did not grasp the recent production situation, and then passively launched the auto market; in Fig 8B, when both $\varphi_1$ and $\varphi_2$ are 0.5, $q_n$ has been fluctuating

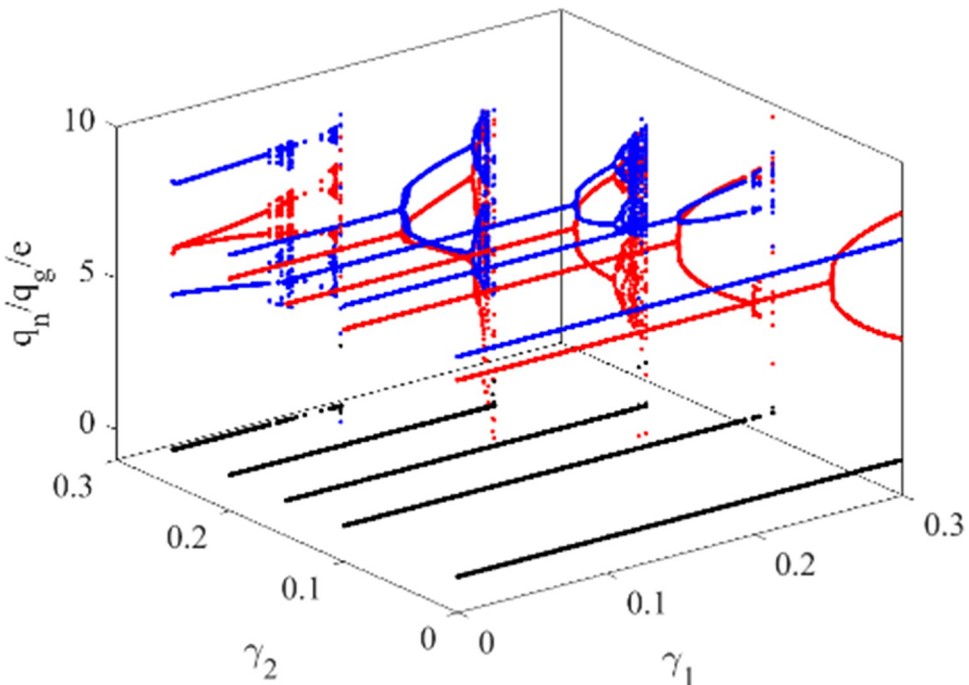

**Fig 7. A bifurcation of the production adjustment parameters of NEV and GV in System (10).**

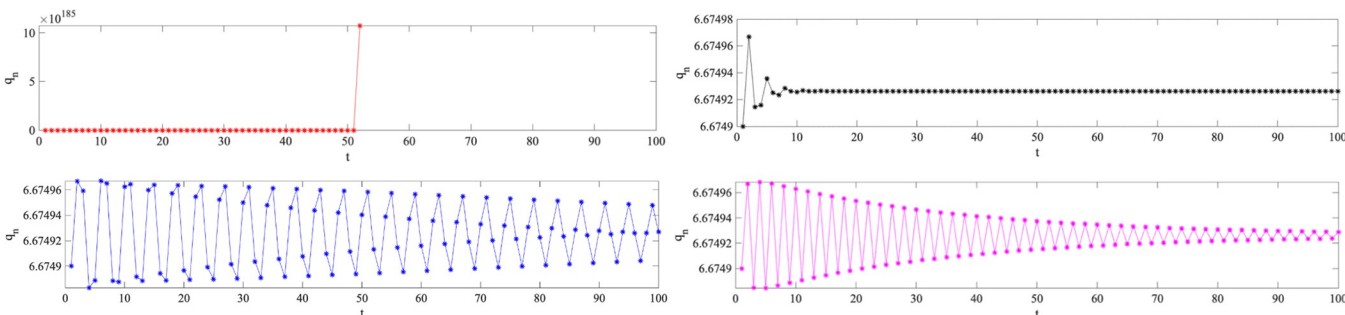

**Fig 8. The time series of NEVs' production for a ($\gamma_1 = 0.1, \gamma_2 = 0.1, \gamma_3 = 1, \varphi_1 = \varphi_2 = 0$); b ($\gamma_1 = 0.1, \gamma_2 = 0.1, \gamma_3 = 1, \varphi_1 = \varphi_2 = 0.5$); c ($\gamma_1 = 0.1, \gamma_2 = 0.1, \gamma_3 = 1, \varphi_1 = \varphi_2 = 0.8$); d ($\gamma_1 = 0.1, \gamma_2 = 0.1, \gamma_3 = 1, \varphi_1 = \varphi_2 = 1$).**

periodically with different amplitude, and the fluctuation amplitude gradually decreases with the increasing of time; in Fig 8C, $q_n$ fluctuates periodically with different amplitude firstly, and has been at a stable state since the 13th cycle; in Fig 8D, similar to Fig 8B, $q_n$ has been fluctuating periodically with different amplitude, but the fluctuation period is more regular. This phenomenon intuitively illustrates the importance of the weight factor to the stability of the auto market, and automakers can adjust the production strategy more significantly, to ensure an increase in profit while controlling the vehicle price.

## 5. Conclusions and management implications

In this paper, the effects of the adjustment of production and carbon reduction, and delay parameters on the stability of nonlinear dynamic system are studied. The simulation results provide a reference for automakers to make reasonable decisions. The main conclusions of the paper are as follows: (1) Within the stability range of the system, the carbon emission reduction adjustment of GVs will not affect the production of NEVs and GVs. Compared with GVs, the production adjustment elasticity of NEVs is more flexible. Therefore, the two automakers should actively respond to the national policy, and GV manufactures should reduce the carbon emissions of their vehicles as much as possible while maintaining the stability of the automobile market, while NEV manufactures can choose to produce more NEVs. (2) As the production and carbon emission reduction adjustment parameters continue to increase, the system of NEVs and GVs will be unstable and eventually enter a chaotic state. Therefore, automakers should make adjustment strategies to avoid adjusting parameters beyond the stability threshold, and the government should also take macro-control measures to avoid shocks in the auto market. The critical points of different states in the bifurcation diagram cannot be visually seen but can be clearly indicated by the LLE. Therefore, the LLE diagram is more suitable to identify the state of the system. (3) When the production adjustment parameters of NEVs and GVs increase at the same time, the system tends to become chaos faster. Because the demand of the auto market is certain, when the vehicle production is too high, there will be an oversupply, then the auto market will be in chaos. Therefore, when making adjustment decisions, automakers should not only consider their own situation, but also consider the strategies of competitors, to maintain the stable operation of the automobile market. (4) Delay production decisions will expand the stability range of the system and increase the stability of the auto market. Moreover, when production weight factors of NEVs and GVs in a certain threshold, the system will move to a stable state faster, which is more conducive to the long-term operation of the automakers.

Based on the above research conclusions, the following management implications and policy recommendations are put forward: (1) When the auto market is chaotic, the decision-making system of the two automakers will enter a disordered state, which will shock the automobile industry. Therefore, it is necessary to analyze and control the chaos of the system to alleviate the industry shock. At the same time, the government should strengthen supervision, create a market environment for sustainable and healthy development, and enhance the vitality of market competition. (2) In addition, delay decisions can expand the stability range of the system, and help automakers to find equilibrium strategies more quickly, which is of great significance for the automaker's profit maximization. From the perspective of managers, small adjustment can keep the system in a stable state, so finding an appropriate weight factor can make the decision more flexible and effective. (3) Under the background of the DC policy, although GV manufactures face great pressure, the adjustment range of carbon emission reduction is very large. Therefore, the government should take the carbon emission reduction of GV manufactures into consideration when making policy adjustments. On one hand, the supervision mechanism should comply with the law of market development and avoid the runaway of the auto market. On the other hand, the government should guide GV manufactures to reduce carbon emission as much as possible, thus promoting the vigorous development of China's auto market.

While this paper provides an in-depth analysis, there are several limitations that warrant mention. Firstly, the research scope should be broadened to consider the presence of plug-in hybrid electric vehicles in the market, transitioning from a duopoly to a more intricate triopoly model. Secondly, future studies should consider a wider range of factors influencing consumer decision-making. This includes not only preferences for low-carbon options and price sensitivity but also essential aspects such as vehicle aesthetics, the quality of after-sales service, and the long-term costs of ownership. Additionally, the carbon emission of GV is assumed to exceed the fuel consumption standard stipulated by the government, and the decision of GV manufactures should be studied after classifying them according to whether they get positive credit. Lastly, although this study concentrates on China's automotive market, it is essential to acknowledge the global shift towards new energy in the automotive industry. Therefore, our future research will aim to explore a more universally applicable approach to facilitate the new energy transformation of the automotive sector across various countries and regions worldwide.

## Author Contributions

**Conceptualization:** Xinying Si, Qiong Wu.

**Formal analysis:** Xinying Si.

**Investigation:** Jinhuan Tang.

**Methodology:** Xinying Si.

**Software:** Xinying Si, Qiong Wu.

**Supervision:** Jinhuan Tang, Xiangchen Li.

**Validation:** Xiangchen Li.

**Visualization:** Xiangchen Li.

**Writing – original draft:** Qiong Wu.

**Writing – review & editing:** Jinhuan Tang, Xinying Si, Qiong Wu.

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
