## [Decision Letter · Decision Letter 0]

23 Sep 2024

PONE-D-23-37726Research on complex dynamic behavior of automakers based on complex system theoryPLOS ONE

Dear Dr. Wu,

Thank you for submitting your manuscript to PLOS ONE. After careful consideration, we feel that it has merit but does not fully meet PLOS ONE’s publication criteria as it currently stands. Therefore, we invite you to submit a revised version of the manuscript that addresses the points raised during the review process.

**ACADEMIC EDITOR: **In line with the reviewer comments I recommend a revision for this manuscript. In particular please look into concerns raised about explaining the terms from dynamical systems theory and time series analysis, as well as expanding the discussion section. Both reviewers have pointed out minor issues with formatting and abbreviations. Please pay careful attention to these. Also, please note that the papers that were suggested by reviewer 1 are optional to cite and citing them will not affect the final outcome of the review process.Please also note that comments by reviewer 2 have been provided in a separate word document.

We look forward to receiving your revised manuscript.

Kind regards,

Sandip Varkey George, PhD

Academic Editor

PLOS ONE

“Jinhuan Tang received each award. The first fund is the National Natural Science Foundation of China, and grant number is  71702112.  The second is Social Science Planning Fund of Liaoning Province under Grant, and grant number is  L19CGL008.”

Reviewers' comments:

Reviewer's Responses to Questions

**Comments to the Author**

1. Is the manuscript technically sound, and do the data support the conclusions?

Reviewer #1: Yes

Reviewer #2: Yes

2. Has the statistical analysis been performed appropriately and rigorously? 

Reviewer #1: N/A

Reviewer #2: Yes

3. Have the authors made all data underlying the findings in their manuscript fully available?

Reviewer #1: Yes

Reviewer #2: Yes

4. Is the manuscript presented in an intelligible fashion and written in standard English?

Reviewer #1: Yes

Reviewer #2: Yes

5. Review Comments to the Author

Reviewer #1: Hello Dear Dr.

This paper addresses on the nonlinear dynamic system to investigate the dynamic adjustment strategies of

production and emission reduction decisions of NEV and GV automakers in a long-term game process under the dual credit (DC) policy using complex system theory.

1. The paper shoud be checked all and corrected according to journal template.

2. The motivation of the work should be further illustrated by introducing more up to date related work. New papers about subject should be reviewed, cited in the introduction section and added to the reference section.

3. In the analysis of the system, Lyapuov or Kaplan-Yorke dimension, the divergence of the system, equilibrium, Symmetry and invariance, Lyapunov exponent can be calculated, analyzed and added to the paper as in the paper entitled with ‘A new hyperchaotic system from T chaotic system: dynamical analysis, circuit implementation, control and synchronization’

4. The space may be left after units.

5. All abbreviations are written clearly when it is used firstly.

6. Any software used for simulations (i.e MATLAB) may need to be cited.

i.e.

(2021), "A new hyperchaotic system from T chaotic system: dynamical analysis, circuit implementation, control and synchronization", Circuit World, https://doi.org/10.1108/CW-09-2020-0223.

Recursive backstepping control of ferroresonant chaotic oscillations consisting between grading capacitor with nonlinear inductance of voltage transformer. Eur. Phys. J. Spec. Top. (2021). https://doi.org/10.1140/epjs/s11734-021-00150-9

Reviewer #2: Dear Authors,

Thank you for submitting your manuscript. Your research on the dynamic behaviors of automakers in the context of China’s dual credit policy (DC) provides significant theoretical insights into the evolving NEV and GV market. In the docx file, I have outlined several strengths and weaknesses of your manuscript, along with suggestions for improvement.

6. PLOS authors have the option to publish the peer review history of their article (what does this mean?). If published, this will include your full peer review and any attached files.

Reviewer #1: No

Reviewer #2: No

---

## [Author Response · Author response to Decision Letter 0]

17 Oct 2024

1. Guided by the valuable suggestions from the reviewers, we have meticulously revised the paper to enhance its academic value and clarity of expression. First, we corrected spelling and grammatical errors to ensure linguistic accuracy. Next, we added detailed explanations of key technical terms to help a broader readership understand our research. Additionally, we refined the language of the article by incorporating abbreviations to make the content more concise. Finally, we optimized the presentation of formulas in the figures to improve their readability and the efficiency of information transfer. We believe these improvements will allow the paper to contribute more significantly to academic discussions, and we look forward to your review and feedback.

2. We have carefully polished the overall language of the article to ensure accuracy and professionalism in our expression. Each sentence has been scrutinized to make the language more concise and fluid while maintaining the rigor of the paper. We believe that these improvements will significantly enhance the quality of the manuscript, better serving the academic community and its readers.

---

## [Decision Letter · Decision Letter 1]

29 Oct 2024

PONE-D-23-37726R1Research on complex dynamic behavior of automakers based on complex system theoryPLOS ONE

Dear Dr. Wu,

Thank you for submitting your manuscript to PLOS ONE. After careful consideration, we feel that it has merit but does not fully meet PLOS ONE’s publication criteria as it currently stands. Therefore, we invite you to submit a revised version of the manuscript that addresses the points raised during the review process.

We look forward to receiving your revised manuscript.

Kind regards,

Feier Chen, Ph.D

Academic Editor

PLOS ONE

Journal Requirements:

Reviewers' comments:

Reviewer's Responses to Questions

**Comments to the Author**

1. If the authors have adequately addressed your comments raised in a previous round of review and you feel that this manuscript is now acceptable for publication, you may indicate that here to bypass the “Comments to the Author” section, enter your conflict of interest statement in the “Confidential to Editor” section, and submit your "Accept" recommendation.

Reviewer #1: (No Response)

Reviewer #2: All comments have been addressed

2. Is the manuscript technically sound, and do the data support the conclusions?

Reviewer #1: Yes

Reviewer #2: Yes

3. Has the statistical analysis been performed appropriately and rigorously? 

Reviewer #1: Yes

Reviewer #2: Yes

4. Have the authors made all data underlying the findings in their manuscript fully available?

Reviewer #1: Yes

Reviewer #2: Yes

5. Is the manuscript presented in an intelligible fashion and written in standard English?

Reviewer #1: Yes

Reviewer #2: Yes

6. Review Comments to the Author

Reviewer #1: Dear Prof,

All comments may not be completely corrected in the revised paper. Please highlight and reply them.

Best Regards.

Reviewer #2: The reviewed article, titled "Research on complex dynamic behavior of automakers based on complex system theory," presents a rigorous analysis of the dynamic behavior of new energy vehicle (NEV) and gasoline vehicle (GV) manufacturers under China's Dual Credit (DC) policy, employing complex systems and nonlinear dynamics theories. Below is a detailed critique based on its strengths and areas for improvement.

Strengths:

Relevance of Topic: The focus on the dynamic behavior of NEV and GV automakers under the DC policy is timely and addresses a significant issue in the context of China's carbon neutrality goals. This is a pertinent study as the global auto industry shifts towards sustainability.

Methodological Rigor: The use of a nonlinear dynamic system and a long-term repeated game model introduces a sophisticated approach to understanding market behavior and the competition between NEV and GV manufacturers. The incorporation of bounded rationality and delay decisions enhances the realism of the model, acknowledging the uncertainties manufacturers face in real-life decision-making.

Comprehensive Theoretical Framework: The paper thoroughly reviews relevant literature on the DC policy and supply chain dynamics, drawing from diverse perspectives to justify the need for the proposed model. The transition from traditional linear models to complex system theory is well-articulated.

Significant Contributions: The paper makes notable contributions, such as constructing a long-term repeated game model specific to the automotive market and incorporating delay decisions to capture the complex dynamics within the system. The use of bifurcation, chaos theory, and Lyapunov exponents in analyzing system stability is a strong point.

Clear Policy Implications: The study provides actionable insights for government regulation and automaker strategies, highlighting how production and emission reduction adjustments can influence the stability of the auto market. This offers practical recommendations for managing market fluctuations and promoting a smoother transition to NEVs.

Conclusion:

The article provides a strong contribution to the literature on the automotive industry’s transformation under environmental policies. Its use of complex systems theory and nonlinear dynamics offers new insights into the stability and chaos in NEV and GV production decisions. However, the practical utility of the model would benefit from further empirical validation, a broader geographical focus, and more accessible presentation of the findings. Nonetheless, it stands as a theoretically rich and methodologically sound contribution to the study of dynamic systems in the context of sustainable automotive policies.

7. PLOS authors have the option to publish the peer review history of their article (what does this mean?). If published, this will include your full peer review and any attached files.

Reviewer #1: No

Reviewer #2: No

---

## [Author Response · Author response to Decision Letter 1]

6 Nov 2024

Dear Editor and reviewers of PLOS ONE：

First and foremost, I would like to express my sincere gratitude for the valuable time and insights you dedicated to our previous submission. We have carefully considered each comment and diligently incorporated the necessary revisions into our manuscript.

Since the reviewers did not provide specific suggestions for revision in this round, we have not prepared individual responses to the reviewers' comments.

We look forward to your further guidance and feedback on the revised manuscript.

Thank you once again for your assistance.

Sincerely. 

Qiong Wu.

---

## [Decision Letter · Decision Letter 2]

12 Nov 2024

PONE-D-23-37726R2Research on complex dynamic behavior of automakers based on complex system theoryPLOS ONE

Dear Dr. Wu,

Thank you for submitting your manuscript to PLOS ONE. After careful consideration, we feel that it has merit but does not fully meet PLOS ONE’s publication criteria as it currently stands. Therefore, we invite you to submit a revised version of the manuscript that addresses the points raised during the review process.

We look forward to receiving your revised manuscript.

Kind regards,

Feier Chen, Ph.D

Academic Editor

PLOS ONE

Journal Requirements:

Reviewers' comments:

Reviewer's Responses to Questions

**Comments to the Author**

1. If the authors have adequately addressed your comments raised in a previous round of review and you feel that this manuscript is now acceptable for publication, you may indicate that here to bypass the “Comments to the Author” section, enter your conflict of interest statement in the “Confidential to Editor” section, and submit your "Accept" recommendation.

Reviewer #1: (No Response)

2. Is the manuscript technically sound, and do the data support the conclusions?

Reviewer #1: Yes

3. Has the statistical analysis been performed appropriately and rigorously? 

Reviewer #1: Yes

4. Have the authors made all data underlying the findings in their manuscript fully available?

Reviewer #1: Yes

5. Is the manuscript presented in an intelligible fashion and written in standard English?

Reviewer #1: Yes

6. Review Comments to the Author

Reviewer #1: Dear Prof,

All comments may not be completely corrected in the revised paper. Please highlight and reply them.

Best Regards.

7. PLOS authors have the option to publish the peer review history of their article (what does this mean?). If published, this will include your full peer review and any attached files.

Reviewer #1: No

---

## [Author Response · Author response to Decision Letter 2]

14 Nov 2024

Dear Editor and reviewers of PLOS ONE

First and foremost, l would like to express my sincere gratitude for the valuable time and insights you dedicated to our previous submission. We have carefully considered each comment and diligently incorporated the necessary revisions into our manuscript.

Since the reviewers did not provide specific suggestions for revision in this round, we have not prepared individual responses to the reviewers' comments.

We look forward to your further guidance and feedback on the revised manuscript.

Thank you once again for your assistance.

Sincerely.

Qiong Wu.

---

## [Decision Letter · Decision Letter 3]

19 Nov 2024

Research on complex dynamic behavior of automakers based on complex system theory

PONE-D-23-37726R3

Dear Dr. Wu,

We’re pleased to inform you that your manuscript has been judged scientifically suitable for publication and will be formally accepted for publication once it meets all outstanding technical requirements.

Kind regards,

Feier Chen, Ph.D

Academic Editor

PLOS ONE

Additional Editor Comments (optional):

Reviewers' comments:

Reviewer's Responses to Questions

**Comments to the Author**

1. If the authors have adequately addressed your comments raised in a previous round of review and you feel that this manuscript is now acceptable for publication, you may indicate that here to bypass the “Comments to the Author” section, enter your conflict of interest statement in the “Confidential to Editor” section, and submit your "Accept" recommendation.

Reviewer #1: (No Response)

2. Is the manuscript technically sound, and do the data support the conclusions?

Reviewer #1: Yes

3. Has the statistical analysis been performed appropriately and rigorously? 

Reviewer #1: Yes

4. Have the authors made all data underlying the findings in their manuscript fully available?

Reviewer #1: Yes

5. Is the manuscript presented in an intelligible fashion and written in standard English?

Reviewer #1: Yes

6. Review Comments to the Author

Reviewer #1: (No Response)

7. PLOS authors have the option to publish the peer review history of their article (what does this mean?). If published, this will include your full peer review and any attached files.

Reviewer #1: No

---

## [Editor Report · Acceptance letter]

28 Nov 2024

PONE-D-23-37726R3 

PLOS ONE

Dear Dr. Wu, 

I'm pleased to inform you that your manuscript has been deemed suitable for publication in PLOS ONE. Congratulations! Your manuscript is now being handed over to our production team.

Kind regards, 

on behalf of

Dr. Feier Chen 

Academic Editor

PLOS ONE